# Diabetic and Hypertensive Disorders Following Miscarriage: A Protocol for Systematic Review and Meta-Analysis

**DOI:** 10.3390/ijerph19148324

**Published:** 2022-07-07

**Authors:** Damien Foo, Jennifer Dunne, Gavin Pereira, Amanuel Gebremedhin, Bereket Duko, Gizachew A. Tessema

**Affiliations:** 1Curtin School of Population Health, Curtin University, Perth, WA 6101, Australia; damien.foo@curtin.edu.au (D.F.); jennifer.dunne1@postgrad.curtin.edu.au (J.D.); gavin.f.pereira@curtin.edu.au (G.P.); a.gebremedhin@curtin.edu.au (A.G.); bereketduko.adema@curtin.edu.au (B.D.); 2enAble Institute, Curtin University, Perth, WA 6101, Australia; 3Centre for Fertility and Health, Norwegian Institute of Public Health, 0473 Oslo, Norway

**Keywords:** miscarriage, diabetes mellitus, hypertension, gestational diabetes, gestational hypertension, eclampsia, pre-eclampsia

## Abstract

(1) Background: Miscarriages occur in approximately 15–25% of all pregnancies. There is limited evidence suggesting an association between history of miscarriage and the development of diabetic and hypertensive disorders in women. This systematic review aims to collate the existing literature and provide up to date epidemiological evidence on the topic. (2) Methods: We will search CINAHL Plus, Ovid/EMBASE, Ovid/MEDLINE, ProQuest, PubMed, Scopus, Web of Science, and Google Scholar, using a combination of medical subject headings, keywords, and search terms, for relevant articles related to the association between miscarriage and the risk of diabetic and hypertensive disorders. Cross-sectional, case–control, nested case–control, case–cohort, and cohort studies published from inception to April 2022 will be included in the search strategy. Three reviewers will independently screen studies and the risk of bias will be assessed using the Joanna Briggs Institute Critical Appraisal tool. Where the data permit, a meta-analysis will be conducted. (3) Results: The results of this systematic review will be submitted to a peer-reviewed journal for publication. (4) Conclusions: The findings of this systematic review will instigate efforts to manage and prevent reproductive, cardiovascular, and metabolic health consequences associated with miscarriages.

## 1. Introduction

Miscarriage, defined as the spontaneous loss of pregnancy prior to fetal viability, is the most common pregnancy complication that occurs in approximately 15–25% of all pregnancies, with an estimated 23 million miscarriages occurring worldwide each year [1,2,3,4]. Miscarriage typically occurs within the first 12 weeks of gestation [3]. The approach to defining miscarriage differs between low- and middle-income countries (i.e., pregnancy loss occurring < 28 weeks of gestation or birthweight < 1000 g) [5] and high-income countries (i.e., pregnancy loss occurring < 20–26 weeks of gestation or birthweight < 400–500 g) [3,6].

Miscarriage, particularly recurrent miscarriage, is a risk marker of future complications during pregnancy and birth, such as placental abruption, stillbirth, preterm birth, and fetal growth restriction [3], and it has also been suggested to be associated with the risk of cardiovascular disease [3]. Diabetes mellitus, hypertension and history of gestational diabetes, gestational hypertension, and pre-eclampsia have also been identified as risk factors of cardiovascular disease [7]. Miscarriage is not recognized as a cardiovascular risk factor in the recommended guidelines by the National Vascular Disease Prevention Alliance (Australia) and the American Heart Association, despite sharing common etiological origins with pre-eclampsia and other obstetric complications [8,9].

Cardiovascular disease shares many risk factors with diabetic and hypertensive disorders, such as type-2 diabetes and hypertension [10,11], and there is growing evidence supporting the association between miscarriage and cardiovascular risk. There has also been an increased interest in the potential impacts of miscarriage on the development of diabetic and hypertensive disorders in women. Recent studies have reported that history of miscarriage is associated with an increased risk of diabetes mellitus [10,11,12,13,14], hypertension [11,12], and gestational diabetes [15,16]. However, these associations are yet to be confirmed. It has been proposed that the biological mechanism for the increased cardiovascular and metabolic risk could be due to oxidative stress and inflammation [15,17,18], which could also contribute to other health outcomes, such as gestational diabetes [15]. An alternative hypothesis may be that exposure to miscarriage could initiate an immunological cascade that could lead to the subsequent development of diabetic and hypertensive disorders, such as type-2 diabetes [13]. However, this association is yet to be confirmed.

The aim of this systematic review will be to systematically search, synthesize, and critically review the current observational evidence for the effect of history of miscarriage on pregnancy- and non-pregnancy-related diabetic and hypertensive disorders in women. The evidence obtained from this review is important to inform families and clinicians regarding reproductive, cardiovascular, and metabolic health, particularly in relation to miscarriage.

## 2. Materials and Methods

This study protocol followed the recommendations by the Preferred Reporting for Systematic Review and Meta-Analysis Protocols (PRISMA-P) [19] and has been registered with the International Prospective Register of Systematic Reviews (CRD42022327689).

### 2.1. Study Characteristics

#### 2.1.1. Population

This systematic review will include women with information on their history of miscarriage and with no prior diabetic or hypertensive co-morbidities before pregnancy. We will not apply any restrictions on studies based on geographic setting and date of publication.

#### 2.1.2. Study Design

This systematic review will include all retrospective or prospective observational studies, including cross-sectional, case–control, nested case–control, case–cohort, and cohort studies, which have evaluated the association between the exposure and outcome(s) of interest.

#### 2.1.3. Exposure

The exposure of interest is history of miscarriage. History of miscarriage is defined as the occurrence of one or more miscarriages from a prior pregnancy. We will investigate whether the number of miscarriages is associated with the outcomes.

#### 2.1.4. Comparator/Control

We will include studies comparing women with a history of miscarriage to women with no history of miscarriage.

#### 2.1.5. Outcomes

The outcomes of interest in this systematic review are diabetic disorders, including diabetes mellitus (i.e., type-1 or type-2) and gestational diabetes, and hypertensive disorders, including essential hypertension and gestational hypertension (e.g., eclampsia and pre-eclampsia).

### 2.2. Data Sources and Search Strategy

We will search for peer-reviewed literature in the following electronic databases: CINAHL Plus, Ovid/EMBASE, Ovid/MEDLINE, ProQuest, PubMed, Scopus, and Web of Science, using a combination of medical subject headings, keywords, and search terms related to the exposure and outcomes (Table 1). We will also search Google Scholar for grey literature. The initial search strategy was developed for Ovid/MEDLINE and will be adapted to each electronic database (Appendix A). We will also search the reference list of the included studies for potentially relevant records not identified by the electronic search.

### 2.3. Eligibility Criteria

#### 2.3.1. Inclusion Criteria

The identified studies need to satisfy the following four criteria to be eligible for inclusion: (1) population criterion includes women of reproductive age with no prior diabetic or hypertensive co-morbidities before pregnancy; (2) study design criterion includes all observational studies; (3) exposure criterion includes studies that investigated history of miscarriage as the primary exposure; and (4) outcome criterion includes studies that investigated at least one diabetic or hypertensive outcome.

#### 2.3.2. Exclusion Criteria

Studies will be excluded based on the following five criteria: (1) non-primary studies including case reports and series, commentaries, editorials, letters to the editors or reviews without original data; (2) studies published in languages other than English; (3) studies that do not evaluate the association between history of miscarriage and at least one diabetic or hypertensive disorder; (4) studies with incomplete information on effect estimates (e.g., missing confidence intervals); and (5) studies that include women with prior diabetic or hypertensive co-morbidities before pregnancy.

### 2.4. Data Management

#### 2.4.1. Study Selection

All unique records identified from the electronic databases will be imported into an Endnote library. In the first stage of the review, one reviewer will screen and review the titles and abstracts of all the identified records retrieved during the searches. A second reviewer will be randomly assigned 20% of the total number of records retrieved during the search and will review the titles and abstracts. In the second stage of the review, two independent reviewers will screen and review the full-texts of these potentially relevant records and decisions will be made regarding inclusion/exclusion. The reason for exclusion will be documented. Studies that are deemed to meet the inclusion criteria will be included in the final review. A third reviewer will resolve any conflicts throughout the review process. In accordance with the Preferred Reporting Items for Systematic Reviews (PRISMA) [20], the selection process will be presented in a flow diagram.

#### 2.4.2. Data Extraction

Two independent reviewers will extract data, from the included studies, using a standardized data collection form, developed to extract information on study design, sample size, geographic location, participant demographics, definition and ascertainment of exposure and outcomes, effect sizes and confidence intervals, and response rate.

#### 2.4.3. Data Synthesis and Analysis

The final review will include a narrative description of the data extracted from the included studies. The findings for each individual study will also be presented in a table. If the data permit and the study design, methodology, and exposure and outcomes are similar enough, we plan to perform a meta-analysis. The statistical heterogeneity between studies will be assessed using the I^2^ statistic [21]. If meta-analysis is not possible due to heterogeneity between the studies, individual effect estimates will be presented in a forest plot, with no attempt of estimating a pooled effect estimate.

#### 2.4.4. Risk of Bias (Quality) Assessment

The quality of the included studies will be assessed by two independent reviewers. The Joanna Briggs Institute Critical Appraisal tool will be used to assess the quality of observational cross-sectional, case–control, nested case–control, case–cohort, and cohort studies [22].

### 2.5. Confidence in Cumulative Evidence

We will use the Grading of Recommendations, Assessment and Evaluations (GRADE) guidelines, developed by the GRADE Working Group, to assess the overall quality of the body of evidence for each outcome evaluated. The GRADE system classifies the quality of evidence into four levels, including very low (very little confidence in the effect estimate and the true effect is likely to be substantially different); low (limited confidence in the effect estimate and the true effect may be substantially different); moderate (moderate confidence in the effect estimate and the true effect is likely to be close to the effect estimate; however, it is possible that it is substantially different); and high (highly confident in the effect estimate and the true effect is close to the effect estimate).

### 2.6. Ethics and Dissemination

Ethics approval is not required as primary data will not be collected. This protocol is guided by the PRISMA-P guidelines [19] and the systematic review will be reported in accordance with the PRISMA guidelines [20].

## 3. Results

The results of this systematic review will be published as a peer-reviewed article.

## 4. Discussion

There have been several etiological arguments that have been proposed to support a biological mechanistic pathway between miscarriage and cardiovascular risk. Recent evidence has suggested the association between miscarriage and diabetic and hypertensive disorders, which may suggest that the development of diabetic and/or hypertensive disorder may be involved in this biological mechanistic pathway.

### Study Limitations

It is possible that the final conclusions of this systematic review may be incomplete due to the limited number of studies, as well as the limited number of participants included in the studies; therefore, it may not be possible to perform a formal meta-analysis. However, we will perform a narrative synthesis to collate and summarize the results. In addition, since there is no standard internationally-defined gestational (or birthweight) boundaries for miscarriage, final conclusions may be limited to how miscarriage was defined.

## 5. Conclusions

This systematic review aims to provide comprehensive evidence of the role of miscarriage on the future risk of diabetic and hypertensive disorders in women. The findings of this review may facilitate cardiometabolic screening programs that target women with a history of miscarriage for early screening.

## Figures and Tables

**Table 1 ijerph-19-08324-t001:** Initial Search Strategy Developed for Ovid/MEDLINE.

	Search Terms
*Exposure*	
Miscarriage	MeSH terms: “Abortion, Spontaneous”, “Abortion, Induced”, or “Fetal Death”. Keywords: [Habitual/Recurrent] Miscarriage, Spontaneous/Induced/Habitual/Recurrent/Incomplete Abortion, Spontaneous/Pregnancy/Infant/Fetal loss, and Fetal Death/Demise.
*Outcomes*	
Diabetes mellitus	MeSH term: “Diabetes Mellitus”. Keywords: [Type-1/Type-2] Diabetes [Mellitus].
Gestational diabetes	MeSH terms: “Diabetes, Gestational” or “Pregnancy in Diabetics”. Keywords: (Diabetes [Mellitus] AND Gestation */Pregnan *).
Hypertension	MeSH term: “Hypertension”. Keywords: (Hyperten *) or (High/Increas*/Elevat* Blood pressure)
Gestational hypertension	MeSH term: “Hypertension, Pregnancy-Induced”. Keywords: ((Hyperten * or High/Increas */Elevat * Blood pressure)) AND (Gestation */Pregnan *) or Eclampsia or Pre-eclampsia

Abbreviations: MeSH, medical subject headings.

## Data Availability

Not applicable.

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
