# Peer review of "Diabetic and Hypertensive Disorders Following Miscarriage: A Protocol for Systematic Review and Meta-Analysis"

_ijerph, 2022, doi:10.3390/ijerph19148324_

Round 1
Reviewer 1 Report
I have to peer review a protocol for systematic review and meta-analysis about the association between history of miscarriage and the development of diabetic and hypertensive disorders in women. The topic is justified because there is limited evidence about the subject and because cardiovascular diseases and diabetes are leading causes of morbidity and mortality.
Systematic reviews and meta-analyses are regarded as the best methods to summarize evidence on the effectiveness of healthcare interventions. The objective of this future systematic review is to inform “families and clinicians regarding reproductive, cardiovascular, and metabolic health, particularly in relation to miscarriage.” Lines 66-68. But the risk of cardiovascular diseases and diabetes associated with miscarriage cannot be modified by preventing the occurrence of a miscarriage!
Lines 40-43 are identical to the text from reference 3 : Quenby, S., Gallos, I. D., Dhillon-Smith, R. K., Podesek, M., Stephenson, M. D., Fisher, J., Brosens, J. J., Brewin, 233 J., Ramhorst, R., Lucas, E. S., McCoy, R. C., Anderson, R., Daher, S., Regan, L., Al-Memar, M., Bourne, T., 234 MacIntyre, D. A., Rai, R., Christiansen, O. B., Sugiura-Ogasawara, M., Odendaal, J., Devall, A. J., Bennett, P. R., 235 Petrou, S., Coomarasamy, A., Miscarriage matters: the epidemiological, physical, psychological, and economic 236 costs of early pregnancy loss. Lancet 2021, 397 (10285), 1658-1667.
Lines 49-51 : the association between miscarriage and cardiovascular risk: I suggest a better explanation because maybe this can be studied for recurrent miscarriage based on etiological mechanisms; I am not sure that one miscarriage – probably due to genetic causes is important to investigate in relation to cardiovascular risk.
I wish you success in your work!
Author Response
|
Editor/Reviewer Comments |
Author’s Response |
Reference page |
|
RESPONSE TO THE REVIEWER 1 |
||
|
“But the risk of cardiovascular diseases and diabetes associated with miscarriage cannot be modified by preventing the occurrence of a miscarriage!” |
We agree with the reviewer that preventing miscarriage may not modify the risk of cardiovascular disease or diabetes mellitus. However, if our study provides evidence of increased risk of diabetes or hypertension (pregnancy related or not), we believe that it will inform help to identify a population group to be targeted for early screening.
We have now modified the “5. Conclusions” section of the protocol and presented as follows: “This systematic review aims to provide comprehensive evidence of the role of miscarriage on the future risk of diabetic and hypertensive disorders in women. The findings of this review may facilitate cardiometabolic screening programs which target women with a history of miscarriage for early screening.” |
Page 5, Lines 204-207 |
|
“Lines 40-43 are identical to the text from reference 3 : Quenby, S., Gallos, I. D., Dhillon-Smith, R. K., Podesek, M., Stephenson, M. D., Fisher, J., Brosens, J. J., Brewin, 233 J., Ramhorst, R., Lucas, E. S., McCoy, R. C., Anderson, R., Daher, S., Regan, L., Al-Memar, M., Bourne, T., 234 MacIntyre, D. A., Rai, R., Christiansen, O. B., Sugiura-Ogasawara, M., Odendaal, J., Devall, A. J., Bennett, P. R., 235 Petrou, S., Coomarasamy, A., Miscarriage matters: the epidemiological, physical, psychological, and economic 236 costs of early pregnancy loss. Lancet 2021, 397 (10285), 1658-1667.” |
We have now revised our statement as follows: “Miscarriage, particularly recurrent miscarriage, is a risk marker of future complications during pregnancy and birth, such as placental abruption, stillbirth, preterm birth, and fetal growth restriction [3], and it has also been suggested to be associated with risk of cardiovascular disease [3].” |
Page 1, Lines 40-43 |
|
“Lines 49-51: the association between miscarriage and cardiovascular risk: I suggest a better explanation because maybe this can be studied for recurrent miscarriage based on etiological mechanisms; I am not sure that one miscarriage – probably due to genetic causes is important to investigate in relation to cardiovascular risk.” |
We agree that the etiological mechanisms of the association between miscarriage and risk of cardiovascular disease is limited. In our systematic review, we will be examining whether the number of miscarriages will be associated with any of the outcomes to be included.
We have now modified the “2.1.3. Methods” section of the protocol and presented as follows: “The exposure of interest is history of miscarriage. History of miscarriage is de-fined as the occurrence of one or more miscarriages from a prior pregnancy. We will investigate whether the number of miscarriages is associated with the outcomes.” |
Page 2, Lines 89-91 |
Reviewer 2 Report
The manuscript is well written. The content is within the scope of the Journal. I enjoyed reading the manuscript by Foo et al., on the effect of history of miscarriage on pregnancy- and non-pregnancy-related diabetic and hypertensive disorders in women. I commend the authors for a good written systematic review and meta-analysis.
Author Response
Thank you very much for reviewing my manuscript and for the positive feedback.
Reviewer 3 Report
I have two questions:
- AGE. Age is the number one risk factor for miscarriage, and obviously is an important risk factor for cardiometabolic disorders. How will the authors deal with the age conundrum? Age-corrected studies only?
- Pre-gestational diabetes. Again, (poorly controlled) pre-gestational diabetes is a known risk factor for miscarriage (including congenital malformations). Some patients who have a miscarriage, may not know they are (pre)-diabetic. How will the authors deal with the issue of undiagnosed pre-gestational (pre)diabetes?
Author Response
|
RESPONSE TO REVIEWER 3 |
|
||
|
I have two questions: “AGE. Age is the number one risk factor for miscarriage, and obviously is an important risk factor for cardiometabolic disorders. How will the authors deal with the age conundrum? Age-corrected studies only?” “Pre-gestational diabetes. Again, (poorly controlled) pre-gestational diabetes is a known risk factor for miscarriage (including congenital malformations). Some patients who have a miscarriage, may not know they are (pre)-diabetic. How will the authors deal with the issue of undiagnosed pre-gestational (pre)diabetes?” |
We agree that maternal age and pre-gestational diabetes are key confounders for the association between miscarriage and cardiovascular risk. As part of our data extraction for the systematic review, we plan to identify these confounding variables adjusted in the primary studies. If we identify any studies that did not adjust for these variables, we will acknowledge this as a limitation of the study. |
|
|